# Measuring the Hubble constant with a sample of kilonovae

Michael W. Coughlin [1,2✉], Sarah Antier[3], Tim Dietrich[4,5], Ryan J. Foley[6], Jack Heinzel[7,8], Mattia Bulla [9], Nelson Christensen[7,8], David A. Coulter[6], Lina Issa[9,10] & Nandita Khetan [11]

Kilonovae produced by the coalescence of compact binaries with at least one neutron star are promising standard sirens for an independent measurement of the Hubble constant ($H_0$). Through their detection via follow-up of gravitational-wave (GW), short gamma-ray bursts (sGRBs) or optical surveys, a large sample of kilonovae (even without GW data) can be used for $H_0$ contraints. Here, we show measurement of $H_0$ using light curves associated with four sGRBs, assuming these are attributable to kilonovae, combined with GW170817. Including a systematic uncertainty on the models that is as large as the statistical ones, we find $H_0 = 73.8^{+6.3}_{-5.8}$ km s$^{-1}$ Mpc$^{-1}$ and $H_0 = 71.2^{+3.2}_{-3.1}$ km s$^{-1}$ Mpc$^{-1}$ for two different kilonova models that are consistent with the local and inverse-distance ladder measurements. For a given model, this measurement is about a factor of 2-3 more precise than the standard-siren measurement for GW170817 using only GWs.

[1] School of Physics and Astronomy, University of Minnesota, Minneapolis, MN 55455, USA. [2] Division of Physics, Math, and Astronomy, California Institute of Technology, Pasadena, CA 91125, USA. [3] APC, UMR 7164, 10 rue Alice Domon et Léonie Duquet, 75205 Paris, France. [4] Institut für Physik und Astronomie, Universität Potsdam, Haus 28, Karl-Liebknecht-Str. 24/25, 14476 Potsdam, Germany. [5] Nikhef, Science Park 105, 1098 XG Amsterdam, The Netherlands. [6] Department of Astronomy and Astrophysics, University of California, Santa Cruz, CA 95064, USA. [7] Artemis, Université Côte d'Azur, Observatoire Côte d'Azur, CNRS, CS 34229, F-06304 Nice Cedex 4, France. [8] Physics and Astronomy, Carleton College, Northfield, MN 55057, USA. [9] Nordita, KTH Royal Institute of Technology and Stockholm University, Roslagstullsbacken 23, SE-106 91 Stockholm, Sweden. [10] Département de Phyisque, Université Paris-Saclay, ENS Paris-Saclay, 91190 Gif-sur-Yvette, France. [11] Gran Sasso Science Institute (GSSI), I-67100 L'Aquila, Italy. ✉email: cough052@umn.edu

Since the discovery of the accelerating expansion rate of the universe[1,2], cosmology surveys have tried to measure the properties of dark energy. One of the most common metrics, type Ia supernovae (SNe), which are standardizable, have been an important tool in this endeavour, with the particular benefit of being detectable throughout a large portion of cosmic time. It has been previously found that the cosmic microwave background (CMB) is consistent with $\Lambda_{CDM}$ cosmology, but predicts a value for $H_0$ in direct tension with other measurements[3]. The redshifts of type Ia SNe in hosts with distances, already determined according to Cepheid variables[4], were used in combination with Hubble Space Telescope imaging[5] to obtain a value $4.4\sigma$ distinct from the Planck Collaboration measurement. It is not yet clear whether this tension is due to the experimental procedures themselves—perhaps rooted in some hidden systematic error—or if it indicates a more exotic physics; additional independent measurements are necessary to assess the true source of the tension.

One of the possible independent measurement methods for $H_0$ connects to the multi-messenger observation of compact binary mergers in which at least one neutron star is present. This approach has been vitalized by the recent combined detection of the neutron star merger (BNS) GW170817[6], GRB 170817A[7,8], and the optical transient AT2017gfo[9], found in the galaxy NGC 4993 12 h after the GWs and GRB. In addition to the resulting insight into the equation of state (EOS) of neutron stars[10] and the formation of heavy elements[11], one of the most exciting results was that of the $H_0$ measurement[12]. This measurement is particularly powerful because GWs are standard sirens[13], which do not rely on a cosmic distance ladder and do not assume any cosmological model as a prior (outside of assuming general relativity is correct). The combination of the distance measurement by the GWs and redshift from the electromagnetic counterpart makes constraints on $H_0$ possible[14]. The distance ladder independent measurement using GWs and the host redshift was $H_0 = 68^{+18}_{-8}$ km/s/Mpc (68.3% highest density posterior interval with a flat-in-log prior)[12]; inclusion of all O2 events reduced this uncertainty to $H_0 = 68^{+14}_{-7}$ km/s/Mpc[15]. Improvements on this measurement using more electromagnetic information, such as high angular resolution imaging of the radio counterparts[16] or information about the internal composition of the NSs[17], are also possible.

Here, we show that the electromagnetic evolution of kilonovae —particularly their decay rate and color evolution—can be compared to theoretical models to determine their intrinsic luminosity, making kilonovae standardizable candles[18,19]. Along with the measured brightnesses, kilonovae can be used to measure cosmological distances. We apply two kilonovae models to sGRB light curves to constrain $H_0$ to $H_0 = 73.8^{+6.3}_{-5.8}$ km s$^{-1}$ Mpc$^{-1}$ and $H_0 = 71.2^{+3.2}_{-3.1}$ km s$^{-1}$ Mpc$^{-1}$ for the two models, improving on what has been achieved so far with GWs alone by about a factor of 2-3.

## Results

**Measuring $H_0$ using kilonovae**. Here, we focus on the kilonova observation happening in coincidence with sGRBs. This type of analysis is particularly prescient given the difficulty of searches for GW counterparts during Advanced LIGO and Advanced Virgo's third observing run (O3)[20]. AT2017gfo, synthesized by the radioactive decay of r-process elements in neutron-rich matter ejected during the merger[21,22], is certainly the best sampled kilonova observation to date. Significant theoretical modeling prior to and after GW170817 has made it possible to study AT2017gfo in great detail, including measurements of the masses, velocities, and compositions of the different ejecta types. These measurements rely on models employing both simplified

semi-analytical descriptions of the observational signatures[23] and modelling using full-radiative transfer simulations[24,25].

In addition to the observation of AT2017gfo, GW170817 was associated with GRB 170817A, which proved that at least some of the observed sGRBs are produced during the merger of compact binaries. This multi-messenger observation revealed the possible connection between kilonovae and sGRBs. For both cases, the GRB is then followed by an afterglow visible in X-rays, optical, and radio for days to months after the initial prompt $\gamma$-ray emission derived from the shock of the jet with the external medium. Our sGRB/kilonova sample follows ref. [26], which combined state-of-the-art afterglow and kilonova models, jointly fitting the observational data to determine whether there was any excess light from a kilonova. The analysis showed light curves consistent with kilonovae in the cases of GRB 150101B[27], GRB 050709[28], GRB 160821B[29], and GRB 060614[30]. Naturally, the error bars on the kilonova parameters are larger for these objects than for GW170817, which have light curves with potentially significant contamination from the afterglow. We refer the reader to ref. [26] for extensive discussions of the photometric data quality and light-curve parameters and modeling. On top of these GRB observations, we will also include measurements from GW170817 (GRB 170817A)[18]. While no spectra of the kilonova excesses exist and X-ray excesses may point to shock heating driving these near-infrared emission[29], kilonovae are one possible (if not even the most likely) interpretation of the excesses. We point out that for the purpose of this article, we assume that the light curves are solely caused by a kilonova emission and neglect the possible contamination due to the sGRB afterglow. While this assumption leads to possible biases if strong sGRB afterglows would contaminate the observed data, adding an sGRB afterglow model on top of the kilonova light curves would increase the dimensionality of the analysis significantly and thus no (or only limited) constraints could be obtained.

The idea is to use techniques borrowed from the type-Ia SNe community to measure distance moduli based on kilonova light curves. We use the light-curve flux and color evolution, which do not depend on the overall luminosity, compared to kilonova models, to predict the luminosity; when combined with the measured brightness, the distance is constrained (see Methods). Here, we develop a model for the intrinsic luminosity of kilonovae based on observables, such that the luminosity can be standardized. Given the potential of multiple components and the change in color depending on the lanthanide fraction, it is useful to use kilonova models to perform the standardization. While it may be possible to standardize the kilonova luminosities based on measured properties, as is done for SN Ia cosmology measurements, in this analysis, we assume that we can use quantities inferred from the light-curve models[31]; this assumption will be testable when a sufficiently large sample of high-quality kilonovae observations are available. In this analysis, we use models from Kasen et al.[24] and Bulla[25]. We note that this analysis uses some of the sampling techniques in the kilonova hypothesis testing and parameter estimation as demonstrated in refs. [10,26], but this is a fundamentally orthogonal exercise to the use of observations in one particular band to standardize the light curves based on measurements from theoretical models. The use of one of the kilonovae such as GW170817 to inform standardization could be used however with unknown systematic errors. While other models for kilonovae exist at this point, e.g., ref. [32], they have been shown to give similar light-curve fits to GW170817, and so the expectation is they would give similar results to those here.

We measure the distances to both GW170817 and the sGRBs in our sample using the posteriors for model parameters and the distributions for the measured parameters from the fit. For the sGRBs, we use two distance estimates based on GW170817 to

inform the standardization; we use GW170817's distance combined with the difference between the computed distance moduli to extract the distance moduli for the sGRBs (see Methods). One powerful aspect of this is that for GW170817 in particular, surface brightness fluctuations (SBF) of the host galaxy NGC 4993 (blue)[33] pin the distance to within 1 Mpc. This requirement is similar to SN Ia measurements, where local distance ladders are required to calibrate the measurement. We also perform a comparison where we use the GW-derived posteriors to anchor the distance distribution, which results in broader but consistent posteriors (see Methods). From there, the distance modulus for each sGRB is solved for, resulting in a distribution of distances.

**$H_0$ constraints**. In addition to the study of GW170817/GRB 170817A/AT2017gfo[18], we compute the corresponding values of $H_0$ for the sGRBs[26]. As described above, we use the Kasen et al.[24] and Bulla[25] models, and we assume systematic error bars with 0.1 mag and 0.25 mag errors for comparison; these are chosen to be similar to photometric errors (0.1 mag) and twice as large (0.2 mag) to establish robustness. This broadens the posterior distributions on the ejecta parameters and these systematic errors also reweight the dependence of the eventual $H_0$ measurement on individual objects. Results for all kilonovae and the combined analysis are listed in Table 1 and Fig. 1. To determine the

posterior distribution for $H_0$, we perform a simultaneous fit for two cosmology models. The first is an empirical model taken to be the following[4]

$$D = \frac{z \times c}{H_0} \left( 1 + \frac{1}{2}(1 - q_0)z - \frac{1}{6}(1 - q_0 - 3q_0^2 + j_0)z^2 + O(z^3) \right).$$

(1)

The second is $\Lambda_{\mathrm{CDM}}$, which depends on $H_0$, $\Omega_m$, and $\Omega_\Lambda$. We checked that both analyses give similar constraints on $H_0$, but do

**Table 1 Summary of $H_0$ results.**

| Kilonova | Kasen—0.25 | Kasen—0.1 | Bulla—0.25 | Bulla—0.1 |
|---|---|---|---|---|
| GW170817 | $75^{+9}_{-8}$ | $78^{+9}_{-8}$ | $75^{+7}_{-7}$ | $75^{+6}_{-6}$ |
| GRB 060614 | $68^{+22}_{-16}$ | $66^{+13}_{-12}$ | $71^{+33}_{-20}$ | $66^{+16}_{-11}$ |
| GRB 150101B | $66^{+40}_{-20}$ | $76^{+39}_{-20}$ | $89^{+54}_{-33}$ | $96^{+51}_{-38}$ |
| GRB 050709 | $78^{+29}_{-19}$ | $63^{+14}_{-11}$ | $62^{+9}_{-7}$ | $61^{+8}_{-6}$ |
| GRB 160821B | $64^{+11}_{-10}$ | $63^{+12}_{-9}$ | $62^{+8}_{-6}$ | $62^{+8}_{-6}$ |
| Combined | $71.9^{+6.4}_{-5.5}$ | $73.8^{+6.3}_{-5.8}$ | $69.9^{+3.6}_{-3.7}$ | $71.2^{+3.2}_{-3.1}$ |

We use units of km s$^{-1}$ Mpc$^{-1}$. Individual rows refer to the GRB/GW observations and individual columns to the Kasen et al. and Bulla et al. model assuming a 0.25 and 0.1 mag systematic uncertainty. We note that the GRB individual $H_0$ measurements use the SBF of the host galaxy NGC 4993[33] to pin the distance.

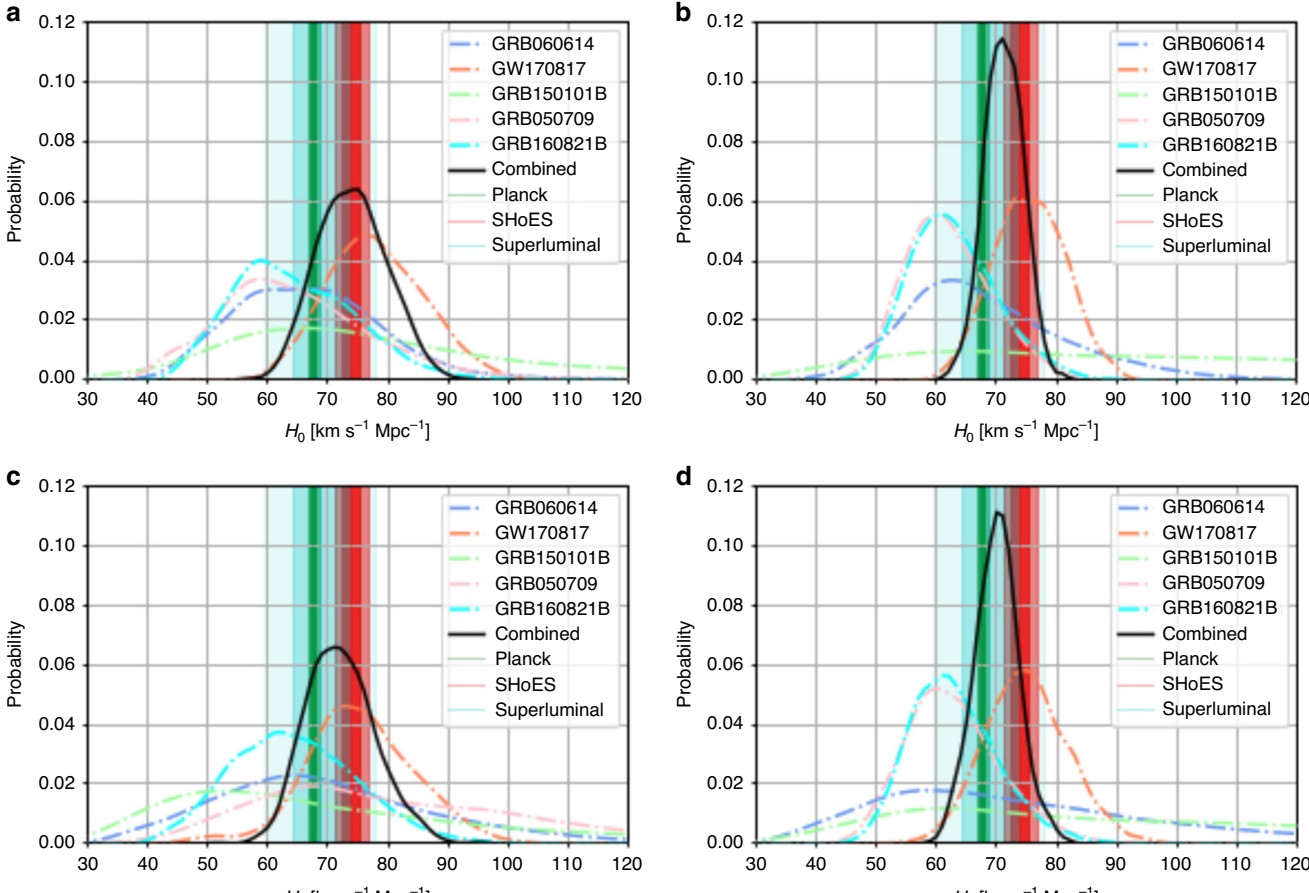

**Fig. 1 Posterior distributions for $H_0$ for individual events.** We show GW170817, GRB 060614, GRB 150101B, GRB 160821B, GRB 050709, and their combined posteriors. Fig. **a** is the Kasen[24] model with 0.1 mag errors, **b** is the Bulla[25] model with 0.1 mag errors, **c** is is the Kasen model with 0.25 mag errors, and **d** is the Bulla model with 0.25 mag errors. The 1- and 2-σ regions determined by the superluminal motion measurement from the radio counterpart (blue)[16], Planck CMB (TT,TE,EE+lowP+lensing) (green)[34] and SHoES Cepheid-SN distance ladder surveys (orange)[5] are also depicted as vertical bands.

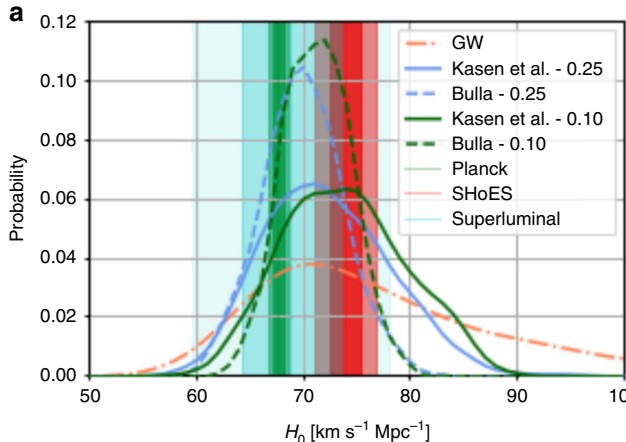
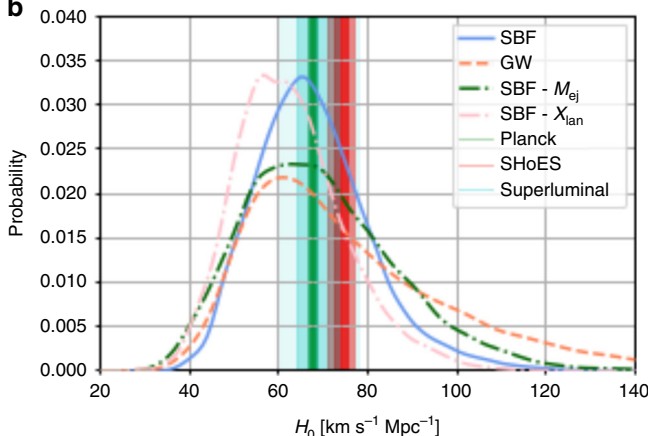

**Fig. 2 Summary of posterior distributions for $H_0$.** In **a**, we show the GW-only analysis for GW170817, in addition to the Kasen[24] and Bulla[25] model analyses with 0.1 mag and 0.25 mag errors from this letter. In addition, the 1- and 2-$\sigma$ regions determined by the superluminal motion measurement from the radio counterpart (blue)[16], Planck CMB (TT,TE,EE+lowP+lensing) (green)[34], and SHoES Cepheid-SN distance ladder surveys (orange)[5] are also depicted as vertical bands. In **b**, we use GRB 060614 and the Kasen[24] model with 0.1 mag errors. In addition to the standard SBF measurement in the letter, we show an analysis where we systematically add 0.1 $M_\odot$ to $M_{ej}$ and 0.5 decades to $X_{lan}$. Finally, we show a distribution where we replace the standard SBF measurement with the distance measurement from the GW170817 high spin posteriors.

not significantly constrain the other model parameters. The systematics from using a particular kilonova model will remain, but the idea is that a sufficient sample of kilonovae will average out variations in the kilonovae. However, the relative consistency between the results of the two models[24,25] yields some confidence that we are still statistics dominated. We caution that the systematic uncertainty could still be significantly larger than what is assumed here, but the expectation is that the difference between the models should be resolved with future observations. The final, combined $H_0$ measurement of our analysis is $H_0 = 73.8^{+6.3}_{-5.8}$ km s$^{-1}$ Mpc$^{-1}$ for the Kasen model and $H_0 = 71.2^{+3.2}_{-3.1}$ km s$^{-1}$ Mpc$^{-1}$ for the Bulla model. These improve on what has been achieved so far with GWs alone by about a factor of 2–3 (see left panel of Fig. 2); the results are consistent with both Planck CMB[34] and SHoES Cepheid-SN distance ladder surveys analyses[5]. We also performed the same analysis without GW170817, due to possible systematic uncertainties from the peculiar velocity of the host; this analysis resulted in both larger error bars than the analysis with GW170817, while still being consistent with it (see Methods).

We can use our results to construct a so-called Hubble–Lemaître Diagram (see Fig. 3), where we plot distance modulus vs. redshift for the observed kilonovae. The main benefit of plotting distance modulus instead of apparent magnitude is that it is independent of the source. For comparison, the green dashed line shows $\Lambda_{CDM}$, showing the consistency in the results. We include a Hubble–Lemaître residual panel to show the error bars. The error bars are, of course, large relative to SNe samples, where $\sigma \sim 0.1$ mag.

## Discussion

We perform a few different tests to assess the systematics of the analysis (see right panel of Fig. 2). The first is where we systematically change the estimated values for the ejecta mass and lanthanide fractions from the light-curve analysis to assess the dependence on those values. Based on ref. [35], we take values of 0.1 $M_\odot$ to add to the ejecta mass and 0.5 decades to add to $X_{lan}$, corresponding to the approximate size of the error bars on those parameters. We show these curves along with the original analysis of GRB 060614 on the right of Fig. 2. While the shift in the derived distribution is clear, in particular for $X_{lan}$, the distributions

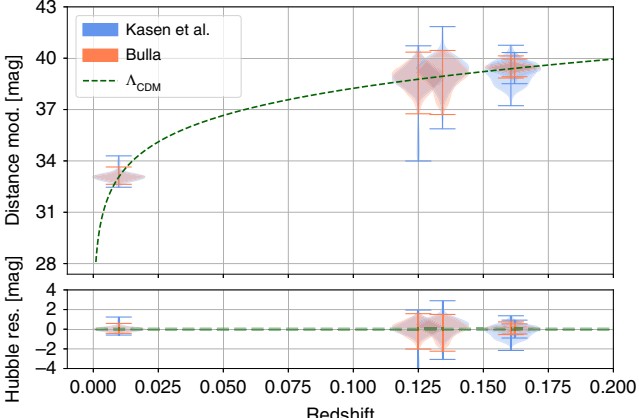

**Fig. 3 Hubble–Lemaître Diagram for the kilonova analysis.** We plot distance modulus vs. redshift. The error bars indicate the extent of the posterior samples. The green dashed line shows $\Lambda_{CDM}$. We include a Hubble–Lemaître residual panel to show the residuals.

still remain consistent with one another; we find distributions of $67^{+13}_{-11}$ for SBF, $67^{+19}_{-15}$ for SBF—$M_{ej}$, and $62^{+13}_{-11}$ for SBF—$X_{lan}$. This indicates that we are still dominated by statistical errors. As a further test, we show a distribution where we replace the standard SBF measurement with the distance measurement from the high spin posteriors presented in ref. [36]. Given the wide posterior distributions, this leads to a broadening of the $H_0$ measurement, but again, the distributions are consistent with one another. In the following, we will use the SBF measurement as the distance anchor for the analysis, but in the future, the GW based distributions may be appropriate to minimize systematics.

These results indicate that continued searches for and the analyses of sGRB afterglows have significant science benefits. In particular, our analysis shows that $H_0$ measurements may be improved given more detections of sGRB afterglows and their peak luminosities, requiring detections as early as possible to be most usable. These objects also have observational challenges, at least those identified by the Fermi Gamma-ray Burst Monitor[37], given the large sky areas that require coverage and the candidate vetting that follows[38]. Observations of more kilonovae will

significantly constrain the kilonova peak luminosity distribution[26], which is clearly a driving force of this analysis. This will likely depend on both whether the original system producing a kilonova is a binary neutron star or neutron star-black hole, in addition to the inclination angle influence on the kilonova characteristics. Kilonovae, in particular, are less bright than SNe Ia, and could be more useful for constraining other distance indicators rather than directly as cosmological probes.

## Methods

**Kilonova analysis**. In addition to the photometric error bars arising from the measured signal from noise, the modeling also has associated errors, which we will add in quadrature. Building upon the first analysis[18], where we employed a Gaussian Process Regression (GPR) based interpolation[39] to create a surrogate model of the model of Kasen et al.[24] for arbitrary ejecta properties[10,35], we also use the model of Bulla[25] for comparison. The idea is that we can use these models to derive constraints on possible kilonova light curves. We chose to have large prior boundaries that allows to describe both, kilonova produced by BNSs and by BHNS systems. For the Kasen et al. model, each light curve depends on the ejecta mass $M_{ej}$, the mass fraction of lanthanides $X_{lan}$, and the ejecta velocity $v_{ej}$. We use flat priors for each parameter covering: $-3 \leq \log_{10}(M_{ej}/M_\odot) \leq -1$, $0 \leq v_{ej} \leq 0.3\ c$, and $-9 \leq \log_{10}(X_{lan}) \leq -1$. For the 2D[25] model, each light curve depends on the ejecta mass $M_{ej}$, the half-opening angle of the lanthanide-rich component $\Phi$ (with $\Phi = 0$ and $\Phi = 90°$ corresponding to one-component lanthanide-free and lanthanide-rich models, respectively) and the observer viewing angle $\theta_{obs}$ (with $\cos\theta_{obs} = 0$ and $\cos\theta_{obs} = 1$ corresponding to a system viewed edge-on and face-on, respectively). We again use flat priors for each parameter covering: $-3 \leq \log_{10}(M_{ej}/M_\odot) \leq -1$, $15° \leq \Phi \leq 30°$, and $0 \leq \theta_{obs} \leq 15$. We restrict $0° \leq \theta_{obs} \leq 15°$ for the sGRB analysis because the viewing angle is much closer to the polar axis than for GW170817[40]. By observing the sGRB, we can assume that we are near or within the opening angle of the sGRB jet, which is taken to be less than 15°[41–44]. Because the distribution of the viewing angles of kilonovae from sGRBs are likely quite anisotropic, we would expect this to create an appearance of changing lanthanide fractions as the viewing angle changed for spherical geometries, such as in the model of Kasen et al.[24]; this could cause a bias in the Hubble Constant measurements using spherical models. Asymmetric models such as that of Bulla[25] overcome this potential issue.

Compared to the models presented in ref. [25], those used here adopt thermalization efficiencies from ref. [45] and estimate the temperature at each time from the mean intensity of the radiation field in each region of the ejecta. In addition, we assume that no mass is located below $v_{min} = 0.025$ c (where $c$ is the speed of light) following ref. [46]. Studying the predictions of two independent models with physical assumptions allows us to estimate the systematic uncertainties of our analysis. More specifically, some of these physical assumptions (e.g., spherical or axial symmetry) may give the false impression of special kilonova properties. Doing this study with multiple different models is therefore critical to reveal such systematics.

In addition to the attempt of providing a measure of the systematic uncertainty, our parameter ranges for both models are very agnostic in the sense that we do not restrict us to particular parameter ranges that are predicted by numerical-relativity simulations[35,47], in fact, the proposed method works for an even larger parameter space, thus, it seems to be very general approach that can be employed for a variety of future events.

**Light curves**. All data presented here were compiled from public sources and collated as presented in ref. [10] for GW170817 and ref. [26] for the remaining SGRBs. For GRB 150101B, data can be found in refs. [27,40], for GRB 050709, data can be found in refs. [28,48–50], for GRB 160821B, data can be found in ref. [29,51], and in GRB 060614, data can be found in refs. [30,52,53]. We remind the reader that for the SGRB analyses, we take the peak $r$-band observation for comparison, as opposed to the $K$-band as used in the GW170817 analysis due to the sparsity of available light curves in that band. We perform the parameter fits using the entire light curves, assuming the light curves are the result of kilonovae. While GW170817 remains the only completely unambiguous kilonova detection, ref. [26] and other analyses, such as refs. [54,55] for GRB 160821B, have indicated that these SGRBs are consistent with having kilonovae present. On the other hand, GRB 060614 in particular has a number of possible interpretations, including a tidal disruption event[56], a WD-NS system[57] or a long GRB[58]. We include it for completeness, but further analysis may require its removal in future analysis.

**$H_0$ analysis**. The underpinning assumption for this paper is that kilonovae can be standardized using models for their luminosity and color evolution, which was first explored in ref. [18]. While in principle this method could be used for other transient types which are more numerous, including core collapse or superluminous supernovae, models for their emission properties are not nearly developed as those for kilonovae. This requires, of course, believable models that we expect encode the evolution of the kilonovae we measure. Metzger[59] showed that the semi-analytic methods of Arnett[60] already reproduce much of the expected physics for these systems, and these models are sufficient for predicting the GW170817 lightcurves (e.g., ref. [61]). For radioactivity models[59], it assumes a power term taken to be $P \propto t^\beta$, and is also dependent on the energy in the system (related to the velocity $v$), the ejecta mass $M$, and the opacity $\kappa$. Under these assumptions, the luminosity as a

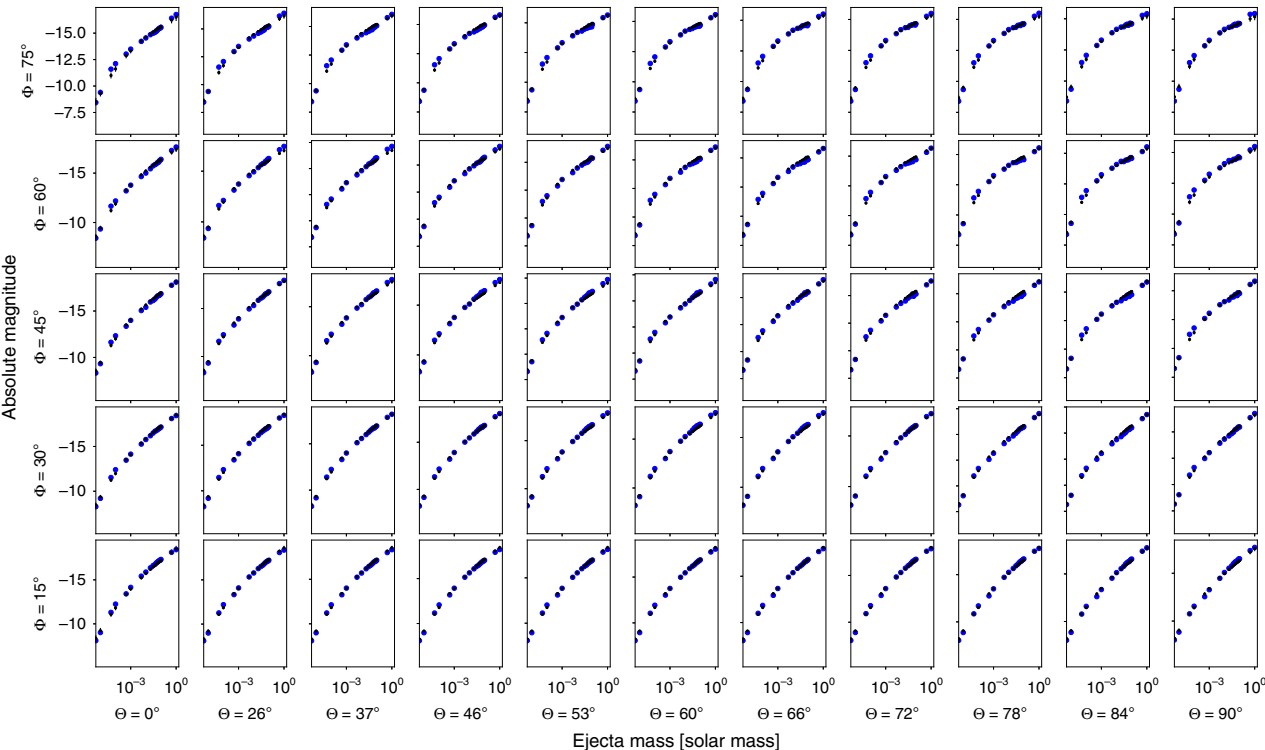

**Fig. 4 Fit of Eq. (3) for the models in ref. [25].** We vary the opening and viewing angles of the employed simulation set in $r$-band. The black points are the fits (with the measured error bar from the Chi-squared) while the blue points are computed directly from the models in ref. [25].

function of time evolves as $\log(L(t)/L_0) \propto -t/\tau$, where $\tau$ is the diffusion timescale $\tau \propto \left(\frac{\kappa M}{v}\right)^{1/2}$ [18]. Numerical models build upon these semi-analytic models with line-based opacities, radiative transport, thermalization efficiency and other parameters to make these models further realistic, although the key point that luminosity depends on these measurable parameters remains.

We now explain explicitly how to derive the standardization. For each model in the simulation set, each of which corresponds to a specific set of intrinsic parameters, we compute the peak magnitude in each passband. We choose $K$-band for the GW170817 analysis when analyzed on its own, given that the transient was observable for the longest in the near-infrared; we chose $r$-band for the sGRB analyses, including in the standardization of GW170817 in the magnitude differences computed below, as it was the band most commonly imaged between the various sGRBs. In this way, we have for both the Kasen et al.[24] model and the Bulla[25] model, a grid of the intrinsic parameters and the peak magnitude associated with the simulation. To map the intrinsic parameters to a peak magnitude, we use a GPR based interpolation (similar to the light-curve interpolation described above)[39].

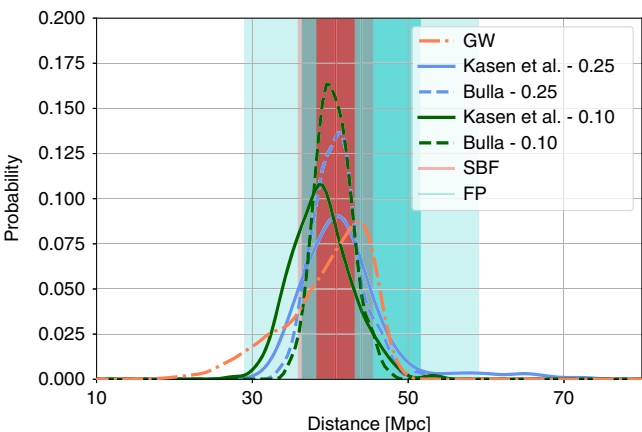

**Fig. 5 Posterior distributions for distance to GW170817.** We show the results of the GW-only analyses (high spin)[36], the Kasen et al.[24] and the Bulla[25] kilonova analysis for both systematic errors assumed. The 1- and 2-$\sigma$ regions determined by the surface brightness fluctuations (SBF) of NGC 4993 (blue)[33] and the Fundamental Plane (FP) of E and S0 galaxies (red)[62] are also depicted as vertical bands.

For the Kasen et al.[24] model, the equation takes the form

$$M_{r=r_{max}} = f(\log_{10}M_{ej}, v_{ej}, \log_{10}X_{lan}) \tag{2}$$

and for the Bulla[25] model,

$$M_{r=r_{max}} = f(\log_{10}M_{ej}, \Phi, \theta_{obs}) \tag{3}$$

where $f$ is a GPR based interpolation and the parameters are inferred quantities based on the light-curve fits. The idea is that these magnitude fits are related to the (observed) apparent magnitudes by the distance modulus $\mu = 5\log_{10}\left(\frac{D}{10pc}\right)$.

$$M = m - \mu. \tag{4}$$

We can evaluate the performance of these fits by comparing the peak $K$-band magnitudes to those predicted as a function of ejecta mass. Figure 4 uses the simulation set for the Bulla[25] model, showing the performance as a function of viewing and opening angle. While the performance tends to be worst for the most extreme viewing and opening angles, the fact that the Gaussian Process estimates errors, which change across the parameter space help to sufficiently reproduce the behavior.

We use the fit of Eqs. (2) and (3) and apply it to the posteriors on the model parameters, as were derived previously for GW170817[10,35] and the sGRBs[26]. Applying directly to the GW170817 posteriors, we show the estimated distances for both models in Fig. 5, consistent with other measurements of the host galaxy, e.g., refs. [33,62,63], and the GW posteriors[36].

To apply the model to the sGRB light curves, we combine Eqs. (2) and (4) as

$$\mu = m - f(\log_{10}M_{ej}, v_{ej}, \log_{10}X_{lan}) \tag{5}$$

and Eqs. (3) and (4) as

$$\mu = m - f(\log_{10}M_{ej}, \Phi, \theta_{obs}). \tag{6}$$

We then take the difference between two observations,

$$\mu_1 - \mu_2 = m_1 - m_2 + f(\log_{10}M_{ej,1}, v_{ej,1}, \log_{10}X_{lan,1}) - f(\log_{10}M_{ej,2}, v_{ej,2}, \log_{10}X_{lan,2}). \tag{7}$$

and

$$\mu_1 - \mu_2 = m_1 - m_2 + f(\log_{10}M_{ej,1}, \Phi_1, \theta_{obs,1}) - f(\log_{10}M_{ej,2}, \Phi_2, \theta_{obs,2}). \tag{8}$$

Combined with either a SBF-based measurement of host galaxy NGC 4993[33] or the GW-derived posteriors to anchor the distance distribution for GW170817, this equation is used to measure the sGRB distance moduli (see Fig. 6 for GRB 060614 as an example). We note that the overall luminosity of the light curve is allowed to vary arbitrarily, which adds a linear offset. But, this offset does not affect the color evolution, which is mostly determining the intrinsic parameter distributions. Therefore, we are able to extract both, the intrinsic parameters and the distance to the source, which is a crucial prerequisite for our study.

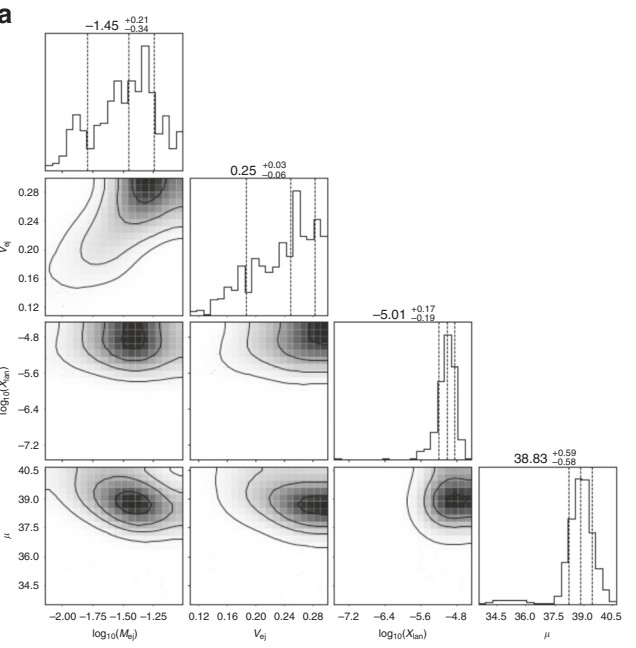
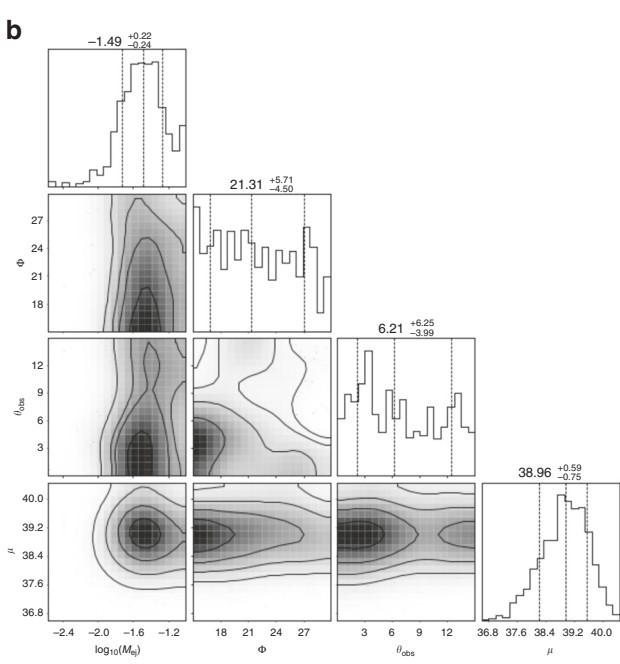

**Fig. 6 Corner plots for the model parameters.** We show them for both Kasen et al.[24] model (**a**) and the Bulla[25] model (**b**) for GRB 060614. We also show the inferred distance modulus in the last column.

**Fig. 7 Posterior distributions for $H_0$ for individual events without GW170817.** We show them for GRB 060614, GRB 150101B, GRB 160821B, GRB 050709, and their combined posteriors are also shown. Fig. **a** is the Kasen[24] model with 0.1 mag errors, **b** is the Bulla[25] model with 0.1 mag errors, **c** is is the Kasen model with 0.25 mag errors, and **d** is the Bulla model with 0.25 mag errors. The 1- and 2-$\sigma$ regions determined by the superluminal motion measurement from the radio counterpart (blue)[16], Planck CMB (TT,TE,EE+lowP+lensing) (green)[34] and SHoES Cepheid-SN distance ladder surveys (orange)[5] are also depicted as vertical bands.

**$H_0$ analysis without GW170817.** In the main text, we included GW170817 when combining the posterior distributions for the Hubble constant. Although the combined analysis is more constraining, the inclusion of GW170817 increases the systematic uncertainties as its H0 measurement depends on the peculiar velocity of the host. This problem will remain for all close-by kilonovae. Due to their distance, the sGRB analysis is not affected nearly as much by the peculiar motion of local galaxies. Removing GW170817's Hubble Constant constraints yields measurements of $H_0 = 71.9^{+8.2}_{-7.7}$ km s$^{-1}$ Mpc$^{-1}$ for the Kasen model and $H_0 = 68.2^{+4.6}_{-4.3}$ km s$^{-1}$ Mpc$^{-1}$ for the Bulla model (see Fig. 7); these measurements both have larger error bars than the analysis with GW170817, while still being consistent with it.

We discussed our prior choices in Section 5. In particular, we choose large prior boundaries that allows to describe both, kilonova produced by BNSs and by BHNS systems, and in particular for ejecta mass, $-3 \leq \log_{10}(M_{ej}/M_\odot) \leq -1$. We can compare this choice to a few other distributions. For example, taking the fit of ref. [35] and applying them to the posteriors of GW170817[6] and GW190425[64], we see a bi-modal distribution, the former peaked near to $\log_{10}(M_{ej}/M_\odot) = -1.3$ and the latter near to $\log_{10}(M_{ej}/M_\odot) = -2.2$; cf. Fig. 8. We also include distributions flat in the component masses with 100 nonparametric EOSs consistent with GW170817 as provided in ref. [65]. This analysis shows that our flat priors cover both of the confirmed events so far, and in this sense, applying flat priors has the significant advantage that a larger range of the parameter space is covered. Further, it reduces systematic uncertainties since no numerical-relativity inferred relations have been applied.

**Choice of ejecta mass priors.** When performing the fits, we rely on the assumption that the standardization is equally valid across the parameter space. Until there are further unambiguous kilonova detections, in particular those that also have GW counterparts, it will be difficult to create more strongly informed

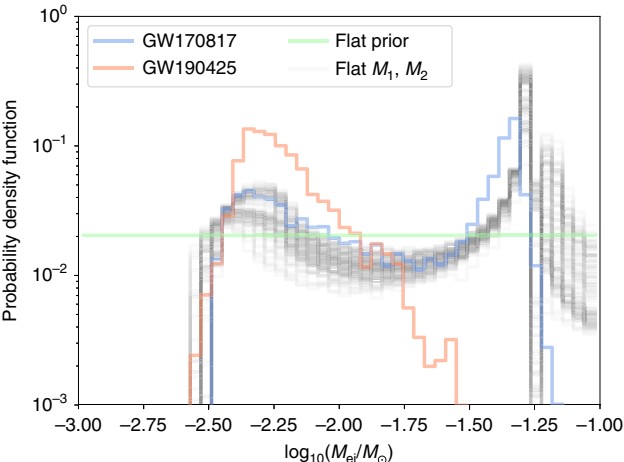

**Fig. 8 Potential prior distributions for $M_{ej}$.** In addition to the flat prior used in this analysis, we show ejecta mass distributions for GW170817[6] and GW190425[64] based on the fit of ref. [35]. We also include distributions flat in the component masses with 100 nonparametric EOSs consistent with GW170817 from ref. [65].

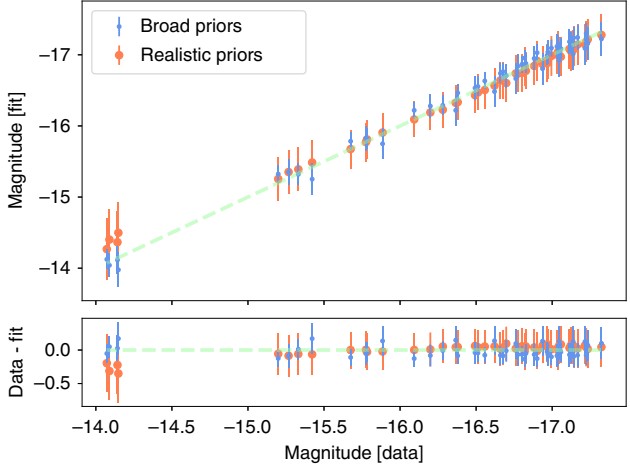

**Fig. 9 Fundamental plane plot.** The top panel shows the fit of Eq. (3) for the models in Bulla[25] model. The Broad Priors values are derived from the entire available grid, covering a range $-6 \le \log_{10}(M_{ej}/M_{\odot}) \le -0$, $15° \le \Phi \le 75°$, and $0° \le \theta_{obs} \le 90°$. The Realistic Priors values are derived from a fraction of the grid drawn from the sGRB informed priors, covering a range $-3 \le \log_{10}(M_{ej}/M_{\odot}) \le -1$, $15° \le \Phi \le 30°$, and $0° \le \theta_{obs} \le 15°$. The bottom panel shows the difference between the computed values from the model and the fits for the simulations analyzed here. The $1 - \sigma$ error bars correspond to those assigned by the Gaussian Process Regression.

priors; for now, doing so would require assumptions on both the distribution of BNS and BHNS masses and an estimate of the NS EOS. It is not so obvious what unphysical regions there are yet, outside of likely very high ejecta masses, which are already excluded by our priors. To evaluate the effect this may have, Figure 9 shows the performance of the fit of Eqs. (3) compared to the Bulla[25] model for both Broad and Realistic priors, i.e., those used in this analysis. Here, the Broad prior values are derived from the entire available grid, covering a range $-6 \le \log_{10}(M_{ej}/M_{\odot}) \le -0$, $15° \le \Phi \le 75°$, and $0 \le \theta_{obs} \le 90$. The Realistic prior values are derived from the prior range used in the sampling, $-3 \le \log_{10}(M_{ej}/M_{\odot}) \le -1$, $15° \le \Phi \le 30°$, and $0° \le \theta_{obs} \le 15°$, which was tuned for the sGRBs. In general, the estimated values from both versions of the fit are consistent with one another and the measured values from the model, and therefore, the effect on the standardization from this perspective is minimal. In the future, if there are regions which are deemed to be disfavored, the priors should be updated to reflect this.

As for assumptions about the ejecta geometry, the current version is generic enough such that it is applicable to both BNS and BHNS cases. However, an improved geometry would likely need to be split into a BNS and BHNS case, which would mean that the standardization for BNS systems could be different from the one for BHNS systems. However, it is currently difficult to assess what regions of the parameter space are favoured for each system and whether these are distinct or overlap. The detection of more kilonovae in the future will help pin down the ejecta geometry and ejecta mass ratio for BNS and BHNS, allowing us to update our priors on $\phi$ and $M_{ej}$ and investigate the possibility of different standardizations.

## Data availability
Upon request, the first author will provide posterior samples from these analyses. All photometric data used in this analysis are publically available from a variety of sources, specified in Methods Section 6 and compiled in https://github.com/mcoughlin/gwemlightcurves/tree/master/lightcurves. Spectral energy distributions for the grid used here will be made available at https://github.com/mbulla/kilonova_models.

## Code availability
The light-curve fitting and Hubble Constant code is available at: https://github.com/mcoughlin/gwemlightcurves.

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

## Acknowledgements

M.W.C. acknowledges support from the National Science Foundation with grant number PHY-2010970. T.D. acknowledges support by the European Union's Horizon 2020 research and innovation program under grant agreement No 749145, BNSmergers. N.C. and J.H. acknowledge support from the National Science Foundation with grant number PHY-1806990. S.A. is supported by the CNES Postdoctoral Fellowship at Laboratoire Astroparticle et Cosmologie. N.C., M.C., and J.H. gratefully acknowledge support from the Observatoire Côte d'Azur, including hospitality for M.C. and J.H. in Summer 2019. The UCSC team is supported in part by NASA grant NNG17PX03C, NSF grant AST-1911206, the Gordon & Betty Moore Foundation, the Heising-Simons Foundation, and by fellowships from the David and Lucile Packard Foundation to R.J.F. D.A.C. acknowledges support from the National Science Foundation Graduate Research Fellowship under Grant DGE1339067.

## Author contributions

M.W.C. conducted the light-curve analysis and was the primary author of the manuscript. M.W.C., S.A., T.D., R.J.F., J.H., M.B., N.C., D.C., L.I., and N.A. contributed to the data analysis procedures and edits to the manuscript.

## Competing interests

The authors declare no competing interests.
