## [Peer Review File · Nature Communications]

Reviewers' comments:

Reviewer #1 (Remarks to the Author):

In the current paper, authors present an interesting method to find the distance to kilonovae without using GW or host galaxy observations. They apply to previous sGRB afterglows and claim to obtain most precise Hubble-Lemaitre constant using these four events.

However, to the best of my knowledge this idea was first proposed by Kashyap+2019 (doi:10.3847/2041-8213/ab543f) to the community using a very similar methodology although only using bolometric luminosity. Please cite them appropriately.

Although the methodology is not novel (presented before in Coughlin et. al. 2019; arXiv:1908.00889), the application of such methodology to past SGRB afterglows is presented for the first time here. Several additional investigations are required to establish the claim presented in the paper.

;General Comments:

(1.) Authors have not provided the details of sampling over parameter space of ejecta mass and velocity, composition and angle. If sampling is done uniformly over all values possible for kilonovae to obtain a mapping between parameter values and peak absolute magnitude in K-band (Methods section), there are two problems that arise because of this--

(1a) Numerical simulations show that not all combinations of parameter values are possible. For example, Radice et. al. 2018 and Coughlin et. al. 2019 show that large ejecta mass and large ejecta velocities are not supported. How will it change your conclusions?

Alternatively, you can adopt the fitting formulae in two references above for your Eqn-2,3 (Methods section) mapping where they properly take into account the non-linearities of numerical simulations.

(1b) Later for some of the GRB, a different band (e.g. r-band) is used for H0 measurement. This seems inconsistent at first thought because the original interpolation was derived in a different band.

(2.) Please note that the use of all bands for testing the kilonova hypothesis and parameter estimation is an orthogonal exercise than the use of observations of one particular band in a standardization method obtained from theoretical model. The use of one of the kilonovae such as GW170817 to inform standardization could be used however with unknown systematic errors.

(3.) Kilonovae could also be observed from NS-BH binary system. Does your methodology work for those systems as well? Please show how and the implications of this fact.

(4.) H_0 measurement is presented here is highly peaked. It appears to be more accurate and in contrast to both Planck and SHOES data. This bold claim is very hard to digest based on sparse afterglows data that are not established definitively as kilonovae.

Furthermore, please quote the results from LIGO GW only and GW-EM measurement of H_0 compared to your results possibly in figure-2 (arXiv:1710.05835,1908.06060; please cite them).

(5.) Authors should note that kilonovae being less brighter than SNe Ia, could be more useful for constraining other distance indicators rather than as cosmological probes.

;Detailed Comments:

'23-24': The process described here and as compared to SNe Ia makes kilonovae standardizable candle and not a standard candle. Please note the difference and provide the correction.

'31-32:' The phrase "...systematic uncertainty as large as statistical..." appears to show that this is the largest error but, it may as well be true that the actual error could be much larger than the both.

'81-82:' I provide here comments on the use of each of the sGRB data. The primary concern is the sparsity of data in K-band which has been used for standardization. I quote references regarding each of the data. Please cite them appropriately.

(*) GRB 150101: K-band observation is only an upper limit at day >1 day which is post-peak.

(*) GRB 050709: Here as well, there are only two K-band observations at day 5.6 and 16.6 which is significantly post-peak.

(* GRB 160821B: There is only one K-band observation available with two upper limits. I believe this gives a very skewed measurement of peak value and decline rate which is very crucial to inferences obtained here. Please cite (10.3847/1538-4357/ab38bb) for their analysis of afterglow observation data for this event.

(* The progenitors of GRB060614 is not settled to be BNS system. It could also be tidal disruption event (doi:10.1086/590899) or, even a WD-NS system (10.1051/0004-6361/200810676). In fact, Xu et.al. 2009 (doi: 10.1088/0004-637X/696/1/971) analyzed the multiband temporal and spectral properties of the two afterglows and found that their afterglows were consistent with the long GRB. I find the use of this event questionable. It could also be an NS-BH merger, in which case please provide details in light of General Comments-(2) above.

'87-89:' I believe the main results in the paper hinges upon this assumption which is very tenuous in the present case.

'93-94:' It was not clear how did you use the color evolution. Please present clearly.

'98-99:' We don't yet know if it's possible to standardize kilonovae; we can only check it provided we have sufficiently large sample of good quality kilonovae observations.

'104-109:' It's not clear why distance to GW is obtained. Which GW and sGRB source are referred to here.

'110-111:' Please show the distance estimate for AT2017gfo using your method of standardization. How does it compare with the distance to the host galaxy?

'117-118:' Again this line is not clear. Please remove if it's not necessary for the main idea of the paper.

'121:' How did the authors come up with the estimate of the systematic error bars? Please provide details in the Methods section.

'line-145-146:' It is important to capture the peak of kilonova during future observation to make this method usable. Multiple kilonovae observation but, post-peak will not be much useful. Kindly make this distinction in the text.

'line-295:' There seems to be some an ad hoc tuning of interpolator whose sensitivity is not shown. Please clarify this.

'281,285' It is not clear whether for "each simulation", you have " a specific set of intrinsic parameters" or, "a grid of parameters". Please elaborate if it's not a contradictory statement.

'312-315:' this statement is not exact. Please provide some quantification.

'322:' Please cite the sources and whether they are public or, protected.

Reviewer #2 (Remarks to the Author):

I have read "Measuring the Hubble Constant with a sample of Kilonovae" by Coughlin et al. The authors demonstrate that the Hubble constant is measured by using the observed kilonovae associated with short GRBs as a standard candle. The authors evaluate the statistical and systematic errors. The Hubble constant obtained here is consistent with those measured by Planck and SH0ES within uncertainties. Once we have more kilonovae associated with a gravitational-wave event as well as with a short GRB, the observations of kilonovae can become a powerful method to probe the cosmological parameters. Therefore I think this paper is worth being published in Nature Communications. However, there are several things in the analysis that are not sufficiently clear to me. I request the authors to address the following concerns.

1. The authors calculate the absolute magnitude at a given band based on the theoretical modelings, which depend on a few model parameters. I think I understand the method of Coughlin et al 17 and Ascenzi et al 19. In this paper, however, it is not clear to me how these parameters are chosen for

each kilonova event without knowing the distance. In other words, how can it be possible that the model parameters are independent of the distance moduli? Are they calibrated by the kilonova in GW170817? I request the authors to describe a bit more details and show the posterior distribution of the model parameters and distance moduli for short GRB kilonovae.

2. This is related to the question 1. The Bulla model depends on the viewing angle. It is not sufficiently clear how the viewing angles for the short GRB kilonovae are chosen. As the authors mention, the association of GRB 170817A with GW suggests that short GRBs are produced by merger. However, this particular GRB is very different from the short GRBs associated with kilonovae. One of likely explanations for this difference is that the viewing angle is different between GRB170817A and the others. For short GRBs, the viewing angle is much closer to the polar axis and therefore kilonovae seen in GRBs are likely to be quite biased toward small viewing angles. I suggest the authors to explicitly mention how the viewing angle is determined and what prior distribution is used. In addition, it would be great if the authors add some discussion about how the unknown distribution of the viewing angles of sGRB kilonovae, which is likely quite anisotropic, causes the bias in the H_0 measurement.

3. The authors include GW170817 to get the combined posterior distribution of the Hubble constant. Although the combined analysis is more constraining H_0 , this adds more systematic uncertainties because the kilonova standard candle method and standard siren method have a different type of systematic errors. Specifically, GW170817 is so close that the H_0 measurement depends on our guess of the peculiar velocity of the host. This problem perhaps will remain even for future events if only close GW-EM events are observed. However, the kilonova standard candle method uses only the distance measurement and does not rely on the peculiar motion of local galaxies. So, I think it will be better not to include the H_0 measurements from GW events in the future (once we have a sufficient number of kilonovae).

I suggest to show the current H_0 posteriors only with short GRB kilonovae. (This is just a suggestion that may be somewhat biased towards my opinion. So please ignore this suggestion if the authors do not want to add).

May 18, 2020

Dear reviewers,

We are grateful for the constructive comments, which have led to a significantly improved manuscript that we resubmit for further evaluation. We gave our best to ensure that the new version is a satisfying response to all raised concerns.

Individual responses to all comments and suggestions are inline below.

Reviewer #1:

In the current paper, authors present an interesting method to find the distance to kilonovae without using GW or host galaxy observations. They apply to previous sGRB afterglows and claim to obtain most precise Hubble-Lemaitre constant using these four events. However, to the best of my knowledge this idea was first proposed by Kashyap+2019 (doi:10.3847/2041-8213/ab543f) to the community using a very similar methodology although only using bolometric luminosity. Please cite them appropriately.

Indeed we (10.1103/PhysRevResearch.2.022006) and Kashyap+2019 developed similar methods independently from each other and both articles appeared almost simultaneously. We added the proper reference to Kashyap+2019 to emphasize this point and thank the referee for pointing this out.

Although the methodology is not novel (presented before in Coughlin et. al. 2019; arXiv:1908.00889), the application of such methodology to past SGRB afterglows is presented for the first time here. Several additional investigations are required to establish the claim presented in the paper.

General Comments:

(1.) Authors have not provided the details of sampling over parameter space of ejecta mass and velocity, composition and angle. If sampling is done uniformly over all values possible for kilonovae to obtain a mapping between parameter values and peak absolute magnitude in K-band (Methods section), there are two problems that arise because of this--

We extended the draft to provide more details about the sampling. The Method section reads now:

“For the Kasen et al. model, each light curve depends on the ejecta mass M_{ej} , the mass fraction of lanthanides X_{lan} , and the ejecta velocity v_{ej} . We use flat priors for each parameter covering: $3 \leq \log_{10}(M_{\text{ej}}/M_{\odot}) \leq -1$, $0 \leq \log_{10}(v_{\text{ej}}/0.3c) \leq -9$, and $3 \leq \log_{10}(X_{\text{lan}}) \leq -1$.

For the 2D (Bul2019) model, each light curve depends on the ejecta mass M_{ej} , the half-opening angle of the lanthanide-rich component Φ (with $\Phi=0$ and $\Phi=90^\circ$ corresponding to one-component lanthanide-free and lanthanide-rich models, respectively) and the observer viewing angle θ_{obs} (with $\cos\theta_{\text{obs}}=0$ and $\cos\theta_{\text{obs}}=1$ corresponding to a system viewed edge-on and face-on, respectively). We again use flat priors for each parameter covering: $3 \leq \log_{10}(M_{\text{ej}}/M_{\odot}) \leq -1$, $15^\circ \leq \Phi \leq 30^\circ$, and $0 < \theta_{\text{obs}} < 15^\circ$. We restrict $0 < \theta_{\text{obs}} < 15^\circ$ for the sGRB analysis because the viewing angle is much closer to the polar axis than for GW170817 (TrRy2018).”

(1a) Numerical simulations show that not all combinations of parameter values are possible. For example, Radice et. al. 2018 and Coughlin et. al. 2019 show that large ejecta mass and large ejecta velocities are not supported. How will it change your conclusions?

The referee is absolutely right that not all combinations of the parameter values are realistic and predicted by numerical-relativity simulations. However, for our analysis, we do not assume any combination of parameters, i.e., we do not make use of any numerical relativity prediction. We use the models of Kasen et al. and Bulla et al. in an agnostic way, in order to be less dependent on a particular model. This means that our analysis is even more general since it indeed holds for all possible parameters, no numerical relativity results enter without our study; we add an additional note to the manuscript stating this fact. In our approach we assume physical ranges broadly consistent with simulations, but larger in order to encompass uncertainties, for the different ejecta parameters and place constraints on these by sampling the multi-D parameter space (including naturally accounting for any degeneracies in the fits).

To make this point more obvious within the manuscript, we added the following discussion:

“Studying the predictions of two independent models with physical assumptions allows us to estimate the systematic uncertainties of our analysis. In addition to the attempt of providing a measure of the systematic uncertainty, our parameter ranges for both models are very agnostic in the sense that we do not restrict us to particular parameter ranges that are predicted by numerical relativity simulations \cite{Radice:2018pdn,CoDi2018b}, in fact, the proposed method works for an even larger parameter space, thus, it seems to be very general approach that can be employed for a variety of future events.”

For the 1D Kasen et al. models, ejecta have no distinctions between different components and are parametrized by ejecta masses, velocities and compositions. For instance, the range of masses goes from $\sim 10^{-3}$ to 10^{-1} Msun and encompasses the range expected from simulations when considering both dynamical and (post-merger) secular ejecta (see e.g. figure 16 of Radice+2018). For the 2D Bulla et al. models, ejecta are divided in two components and parameterized by the total ejecta mass, the extension of the two components (ϕ parameter) and the viewing angle. These two components are meant to represent generic “blue” (low-opacity) and “red” (high-opacity) components and have not necessarily a 1-to-1 relation with ejecta components released during the merger (but see response to point 3 for examples associating ‘red’ components to ‘dynamical’ ejecta). An example of this for GRB060614 is shown in Figure 6, with ejecta masses (\log_{10}) of $-1.45 \pm \sim 0.25$ in the Kasen et al. case and $-1.49 \pm \sim 0.2$ in the Bulla case, consistent with expectations based on the ejecta mass fits.

Alternatively, you can adopt the fitting formulae in two references above for your Eqn-2,3 (Methods section) mapping where they properly take into account the non-linearities of numerical simulations.

The referee is correct that it would be possible to use the formulas of Radice et al., 2018 and Coughlin et al., 2019 to sample on the binary properties and map the binary properties to priors on the ejecta mass properties that are then used for the parameter estimation. We have added to the manuscript a new section in the methods part: *“We discussed our prior choices in Section~\ref{sec:kilonova}. In particular, we choose large prior boundaries that allows to describe both, kilonova produced by BNSs and by BHNS systems, and in particular for ejecta mass, $-3 \leq \log_{10} (M_{\text{ej}}/M_{\odot}) \leq -1$. We can compare this choice to a few other distributions. For example, taking the fit of Ref.~\cite{CoDi2018b} and applying them to the posteriors of GW170817\cite{AbEA2017b} and GW190425\cite{AbEA2019_GW190425}, we see a bi-modal distribution, the former peaked near to $\log_{10} (M_{\text{ej}}/M_{\odot})=-1.3$ and the latter near to $\log_{10} (M_{\text{ej}}/M_{\odot})=-2.2$; cf.~Fig.~\ref{fig:mejprior}. We also include distributions flat in the component masses with $\$100\$ non-$*

parametric EOSs consistent with GW170817 as provided in Ref.~\cite{LaRe2019}. This analysis shows that our flat priors cover both of the confirmed events so far, and in this sense, applying flat priors has the significant advantage that a larger range of the parameter space is covered. Further, it reduces systematic uncertainties since no numerical-relativity inferred relations have been applied.”

In addition, we also added Figure 9 to the draft.

Figure 9: Potential prior distributions for M_{ej} . In addition to the flat prior used in this analysis, we show ejecta mass distributions for GW170817⁸ and GW190425⁷³ based on the fit of Ref. ³⁹. We also include distributions flat in the component masses with 100 non-parametric equation of states consistent with GW170817 as provided in Ref. ⁷⁴.

(1b) Later for some of the GRB, a different band (e.g. r-band) is used for H0 measurement. This seems inconsistent at first thought because the original interpolation was derived in a different band.

We have clarified in the text that r-band is used for all of the sGRB analyses. This is caused by the fact that, as pointed out by the referee, very few sGRB’s have K-band observations. Further details are given in the answers below.

(2.) Please note that the use of all bands for testing the kilonova hypothesis and parameter estimation is an orthogonal exercise than the use of observations of one particular band in a standardization method obtained from theoretical model. The use of one of the kilonovae such as GW170817 to inform standardization could be used however with unknown systematic errors.

To address this, we have added to the text: “We note that this analysis uses some of the sampling techniques in the kilonova hypothesis testing and parameter estimation as demonstrated in Refs.~\cite{CoDi2018,AsCo2018}, but this is a fundamentally orthogonal exercise to the use of observations in one particular band to standardize the light curves based on measurements from theoretical models.”

(3.) Kilonovae could also be observed from NS-BH binary system. Does your methodology work for those systems as well? Please show how and the implications of this fact.

This point is related to point 1a and we hope that our answers provided already a better insight. The models from Kasen et al. and Bulla et al. are agnostic to the specific binary systems and properties. That is, they are generic enough to be used to model both BNS and NSBH systems. In fact, the ejecta parameters are sampled in wide-enough ranges to encompass the values expected from numerical-relativity simulations of BNSs and NSBHs. For instance, one of the largest discriminant between BNS and NSBH mergers is the amount of mass ejected on dynamical timescales. While dynamical ejecta are only a few times 10^{-3} Msun in BNS (see e.g. Radice et al. 2018, figure 16), larger values are found in some NSBH mergers (10^{-2} to 10^{-1} Msun, e.g. Krueger & Foucart 2020). All these values are included in the range adopted for both models. For example, a dynamical ejecta that is lanthanide-rich, around the equatorial plane and with a mass of 0.05 Msun (more typical for NSBH systems) would be captured by the Bulla et al. models with the following (not unique) choice of parameters: $Mej_{tot} = 0.1$ Msun, $\phi = 30$ deg \rightarrow $Mej_{red/equatorial} = Mej_{tot} \times \cos(90-\phi) = 0.05$ Msun. Similarly, a dynamical ejecta of 0.0025 Msun (more typical for BNS systems) would be captured by the Bulla et al. models with the following (not unique) choice of parameters: $Mej_{tot} = 0.01$ Msun, $\phi = 15$ deg \rightarrow $Mej_{red/equatorial} = Mej_{tot} \times \cos(90-\phi) \sim 0.0026$ Msun.

(4.) H_0 measurement is presented here is highly peaked. It appears to be more accurate and in contrast to both Planck and SHOES data. This bold claim is very hard to digest based on sparse afterglows data that are not established definitively as kilonovae.

Furthermore, please quote the results from LIGO GW only and GW-EM measurement of H_0 compared to your results possibly in figure-2 (arXiv:1710.05835,1908.06060; please cite them).

Both GW papers are now cited in the introduction. We have added a dedicated section on the data sets used in order to discuss the afterglow data. In addition, the impression that our results are more accurate than Planck or SHOES data is incorrect. The obtained results, based on the existing dataset, are not accurate enough to distinguish between the Planck and SHOES data. We have added this explicitly to the draft: *“the results are consistent with both Planck CMB (cite{Ade2015}) and SHoES Cepheid-SN distance ladder surveys analyses (cite{RiCa2019}).”*

(5.) Authors should note that kilonovae being less brighter than SNe Ia, could be more useful for constraining other distance indicators rather than as cosmological probes.

We acknowledge this comment and extended our manuscript with: *“Kilonovae, in particular, are less bright than SNe Ia, and could be more useful for constraining other distance indicators rather than directly as cosmological probes.”*

Detailed Comments:

'23-24': The process described here and as compared to SNe Ia makes kilonovae standardizable candle and not a standard candle. Please note the difference and provide the correction.

This inaccuracy has been fixed in the revised version.

'31-32:' The phrase "...systematic uncertainty as large as statistical..." appears to show that this is the largest error but, it may as well be true that the actual error could be much larger than the both.

We have added: *“We caution that the systematic uncertainty could still be significantly larger than what is assumed here.”*

'81-82:' I provide here comments on the use of each of the sGRB data. The primary concern is the sparsity of data in K-band which has been used for standardization. I quote references regarding each of the data. Please cite them appropriately.

(*) GRB 150101: K-band observation is only an upper limit at day >1 day which is post-peak.

(*) GRB 050709: Here as well, there are only two K-band observations at day 5.6 and 16.6 which is significantly post-peak.

(*) GRB 160821B: There is only one K-band observation available with two upper limits. I believe this gives a very skewed measurement of peak value and decline rate which is very crucial to inferences obtained here. Please cite (10.3847/1538-4357/ab38bb) for their analysis of afterglow observation data for this event.

(*) The progenitors of GRB060614 is not settled to be BNS system. It could also be tidal disruption event (doi:10.1086/590899) or, even a WD-NS system (10.1051/0004-6361/200810676). In fact, Xu et.al. 2009 (doi: 10.1088/0004-637X/696/1/971) analyzed the multiband temporal and spectral properties of the two afterglows and found that their afterglows were consistent with the long GRB. I find the use of this event questionable. It could also be an NS-BH merger, in which case please provide details in light of General Comments-(2) above.

We have added a section (Lightcurves) to the methods discussing the source of the data and remind that the r-band observations are used to standardize the SGRBs due to the sparsity of K-band data, and caveats about GRB060614's source in particular; exactly as pointed out by the referee. This section reads: *"All data presented here was compiled from public sources and collated as presented in Ref.~\cite{CoDi2018} for GW170817 and Ref.~\cite{AsCo2018} for the remaining SGRBs. For GRB150101B, data can be found in Refs.~\cite{FoMa2016,TrRy2018}, for GRB050709, data can be found in Refs.~\cite{FoFr2005,Hjorth2005,Covino2006,Jin2016}, for GRB160821B, data can be found in Ref.~\cite{KaKoLau2017,JinWang2018}, and in GRB060614, data can be found in Refs.~\cite{ZhZh2007,JinLi2015,YaJi2015}. We remind the reader that for the SGRB analyses, we take the peak r -band observation for comparison, as opposed to the K -band as used in the GW170817 due to the sparsity of available light curves in that band. We perform the parameter fits using the entire light curves, assuming the light curves are the result of kilonovae. While GW170817 remains the only completely unambiguous kilonova detection, Ref.~\cite{AsCo2018} and other analyses, such as Refs.~\cite{LaTa2019,Rossi2019} for GRB160821B, have indicated that these SGRBs are consistent with having kilonovae present. On the other hand, GRB060614 in particular has a number of possible interpretations, including a tidal disruption event\cite{LuHu2008}, a WD-NS system \cite{CaBe2009} or a long GRB\cite{XuSt2009}. We include it for completeness, but further analysis may require its removal in future analysis."*

'87-89:' I believe the main results in the paper hinges upon this assumption which is very tenuous in the present case.

We have done our best to state the caveats of this as clearly as we can, e.g. in the abstract (Line 32): "Here, we use light curves associated with four short gamma-ray bursts, discovered at a redshift of $z \sim 0.1$ (GRB060614, GRB150101B, GRB160821B and GRB050709) and, assuming these are attributable to kilonovae, combine them with GW170817 to measure H_0 ", line 92, "We point out that for the purpose of this article, we assume that the light curves are solely caused by a kilonova emission and neglect the possible contamination due to the sGRB afterglow", and line 311: "We perform the parameter fits using the entire light curves, assuming the light curves are the result of kilonovae."

'93-94:' **It was not clear how did you use the color evolution. Please present clearly.**

We have extended this by adding: "compared to kilonova models" in this sentence to make it more clear, i.e., *"We use the light curve flux and color evolution, which do not depend on the overall luminosity, compared to kilonova models, to predict the luminosity; when combined with the measured brightness, the distance is constrained (see Methods)."* This should make the rest of the paragraph more understandable based on this sentence.

'98-99:' We don't yet know if it's possible to standardize kilonovae; we can only check it provided we have sufficiently large sample of good quality kilonovae observations.

We have softened the statements that we made and in addition added the following caveat: *“this assumption will be testable when a sufficiently large sample of high-quality kilonovae observations are available.”*

'104-109:' It's not clear why distance to GW is obtained. Which GW and sGRB source are referred to here.

We now state: *“For the sGRBs, we use two distance estimates based on GW170817 to inform the standardization; we use GW170817's distance combined with the difference between the computed distance moduli to extract the distance moduli for the sGRBs (see Methods).”*

'110-111:' Please show the distance estimate for AT2017gfo using your method of standardization. How does it compare with the distance to the host galaxy?

We have added a new Figure (Fig 4) to the manuscript.

Figure 4: Posterior distributions for distance to GW170817, where the results of the GW-only analyses (high spin)⁶⁴, the Kasen et al.²³ and the Bulla²⁴ kilonova analysis for both systematic errors assumed. The 1- and 2- σ regions determined by the surface brightness fluctuations (SBF) of NGC4993 (blue)³³ and the Fundamental Plane (FP) of E and S0 galaxies (red)⁶⁶ are also depicted as vertical bands.

Furthermore, we added additional text to the main text body: *“We use the fit of Eqs. $\sim \text{eqref{eq:fit_inferred_kasen}}$ and $\sim \text{eqref{eq:fit_inferred_bulla}}$ and apply it to the posteriors on the model parameters, as were derived previously for GW170817^{\cite{CoDi2018,CoDi2018b}} and the sGRBs^{\cite{AsCo2018}}.*

Applying directly to the GW170817 posteriors, we show the estimated distances for both models in Figure^{\ref{fig:distance}}}, consistent with other measurements of the host galaxy, e.g. ^{\cite{HjLe2017,LeLy2017,CaJe2018}}, and the gravitational-wave posteriors^{\cite{AbEA2018}}.”

'117-118:' Again this line is not clear. Please remove if it's not necessary for the main idea of the paper.

We removed the line from the manuscript

'121:' How did the authors come up with the estimate of the systematic error bars? Please provide details in the Methods section

We have added: *“these are chosen to be similar to photometric errors (0.1 mag) and twice as large (0.2 mag) to establish robustness.”*

'line-145-146:' It is important to capture the peak of kilonova during future observation to make this method usable. Multiple kilonovae observation but, post-peak will not be much useful. Kindly make this distinction in the text.

This now reads: *“In particular, our analysis shows that H_0 measurements may be improved given more detections of sGRB afterglows and their peak luminosities, requiring detections as early as possible to be most usable.”*

'line-295:' There seems to be some an ad hoc tuning of interpolator whose sensitivity is not shown. Please clarify this.

This now reads: *“the fact that the Gaussian Process estimates errors which change across the parameter space help to sufficiently reproduce the behavior...”*

'281,285' It is not clear whether for "each simulation", you have " a specific set of intrinsic parameters" or, "a grid of parameters". Please elaborate if it's not a contradictory statement.

We have added that it is for each “model” instead of each simulation. It now reads: *“For each model in the simulation set, each of which corresponds to a specific set of intrinsic parameters, we compute the peak magnitude in each passband.”*

'312-315:' this statement is not exact. Please provide some quantification.

We have added the numerical values for the systematics checks. In particular, we have added: *“Removing GW170817’s Hubble Constant constraints yields measurements of $H_0 = 71.9^{+8.2}_{-7.7}$ km s⁻¹ Mpc⁻¹ for the Kasen model and $H_0 = 68.2^{+4.6}_{-4.3}$ km s⁻¹ Mpc⁻¹ for the Bulla model; these measurements both have larger error bars than the analysis with GW170817, while still being consistent with it.”*

'322:' Please cite the sources and whether they are public or, protected.

The new section in the methods now gives the citations and we also include a link to the repository where the photometry is compiled. Specifically, we added: *“All data presented here was compiled from public sources and collated as presented in Ref.~\cite{CoDi2018} for GW170817 and Ref.~\cite{AsCo2018} for the remaining SGRBs. For GRB150101B, data can be found in Refs.~\cite{FoMa2016,TrRy2018}, for GRB050709, data can be found in Refs.~\cite{FoFr2005,Hjorth2005,Covino2006,Jin2016}, for GRB160821B, data can be found in Ref.~\cite{KaKoLau2017,JinWang2018}, and in GRB060614, data can be found in Refs.~\cite{ZhZh2007,JinLi2015,Yaji2015}.”* and *“All photometric data used in this analysis is publically available from a variety of sources, specified in Methods Section~\ref{sec:lightcurves} and compiled in [\url{https://github.com/mcoughlin/gwemlightcurves/tree/master/lightcurves}](https://github.com/mcoughlin/gwemlightcurves/tree/master/lightcurves).”*

Reviewer #2 (Remarks to the Author):

I have read “Measuring the Hubble Constant with a sample of Kilonovae” by Coughlin et al. The authors demonstrate that the Hubble constant is measured by using the observed kilonovae associated with short GRBs as a standard candle. The authors evaluate the statistical and systematic errors. The Hubble constant obtained here is consistent with those measured by Planck and SHOES within uncertainties. Once we have more kilonovae associated with a gravitational-wave event as well as with a

short GRB, the observations of kilonovae can become a powerful method to prove the cosmological parameters. Therefore I think this paper is worth being published in Nature Communications. However, there are several things in the analysis that are not sufficiently clear to me. I request the authors to address the following concerns.

We are grateful for this positive feedback and hope that final concerns are clarified by the answers given below.

1. The authors calculate the absolute magnitude at a given band based on the theoretical modelings, which depend on a few model parameters. I think I understand the method of Couglin et al 17 and Ascenzi et al 19. In this paper, however, it is not clear to me how these parameters are chosen for each kilonova event without knowing the distance. In other words, how can it be possible that the model parameters are independent of the distance moduli? Are they calibrated by the kilonova in GW170817? I request the authors to describe a bit more details and show the posterior distribution of the model parameters and distance moduli for short GRB kilonovae.

We have clarified this by extending the main text: *“We note that the overall luminosity of the light curve is allowed to vary arbitrarily, which adds a linear offset. But, this offset does not affect the color evolution, which is mostly determining the intrinsic parameter distributions. Therefore, we are able to extract both, the intrinsic parameters and the distance to the source, which is a crucial prerequisite for our study.”*

We present an example of the intrinsic parameters and the distance modulus for GRB060614 in Figure 6 that we added to the manuscript for better clarification and illustration.

Figure 6: The corner plots for the model parameters for both Kasen et al.²³ model (left) and the Bulla²⁴ model (right) for GRB060614. We also show the inferred distance modulus in the last column.

2. This is related to the question 1. The Bulla model depends on the viewing angle. It is not sufficiently clear how the viewing angles for the short GRB kilonovae are chosen. As the authors mention, the association of GRB 170817A with GW suggests that short GRBs are produced by merger. However, this particular GRB is very different from the short GRBs associated with kilonovae. One of likely explanations for this difference is that the viewing angle is different between GRB170817A and the

others. For short GRBs, the viewing angle is much closer to the polar axis and therefore kilonovae seen in GRBs are likely to be quite biased toward small viewing angles. I suggest the authors to explicitly mention how the viewing angle is determined and what prior distribution is used. In addition, it would be great if the authors add some discussion about how the unknown distribution of the viewing angles of sGRB kilonovae, which is likely quite anisotropic, causes the bias in the H_0 measurement.

We have added to the text: *“For the sGRB analysis, because the viewing angle is much closer to the polar axis, we restrict $0 < \theta_{\text{obs}} < 15^\circ$. Because the distribution of the viewing angles of kilonovae from sGRBs are likely quite anisotropic, we would expect this to create an appearance of changing lanthanide fractions as the viewing angle changed for spherical geometries, such as in the model of Kasen et al. (KasMe2017); this could cause a bias in the Hubble Constant measurements using spherical models. Asymmetric models such as that of Bulla (Bul2019) overcome this potential issue.”*

We also have extended text in the methods section detailing the form of the priors for the sampling, in particular, they are taken to be flat for each parameter. For example, the priors are uniform in the bounds $-3 \leq \log_{10}(M_{\text{ej}}/M_{\odot}) \leq -1$, $15^\circ \leq \Phi \leq 30^\circ$, and $0 < \theta_{\text{obs}} < 15^\circ$, for the Bulla models.

3. The authors include GW170817 to get the combined posterior distribution of the Hubble constant. Although the combined analysis is more constraining H_0 , this adds more systematic uncertainties because the kilonova standard candle method and standard siren method have a different type of systematic errors. Specifically, GW170817 is so close that the H_0 measurement depends on our guess of the peculiar velocity of the host. This problem perhaps will remain even for future events if only close GW-EM events are observed. However, the kilonova standard candle method uses only the distance measurement and does not rely on the peculiar motion of local galaxies. So, I think it will be better not to include the H_0 measurements from GW events in the future (once we have a sufficient number of kilonovae).

I suggest to show the current H_0 posteriors only with short GRB kilonovae. (This is just a suggestion that may be somewhat biased towards my opinion. So please ignore this suggestion if the authors do not want to add).

We agree that there is some potential for bias with the peculiar velocities, although we expect that marginalizing over reasonable values should have addressed this in the analysis. One can hope that there are more BN-NS kilonovae in the future, bringing the distances potentially significantly further out to limit the effect of peculiar velocities. We have added a separate Section to the Methods where we include a summary of both this discussion and the relevant figures without the GW170817 result included. In particular, we added: *“In the main text, we included GW170817 when combining the posterior distributions for the Hubble constant. Although the combined analysis is more constraining, the inclusion of GW170817 increases the systematic uncertainties as its H_0 measurement depends on the peculiar velocity of the host. This problem will remain for all close-by kilonovae. Due to their distance, the sGRB analysis is not affected nearly as much by the peculiar motion of local galaxies. Removing GW170817's Hubble Constant constraints yields measurements of $H_0 = 71.9^{+8.2}_{-7.7} \text{ km s}^{-1} \text{ Mpc}^{-1}$ for the Kasen model and $H_0 = 68.2^{+4.6}_{-4.3} \text{ km s}^{-1} \text{ Mpc}^{-1}$ for the Bulla model; these measurements both have larger error bars than the analysis with GW170817, while still being consistent with it.”* in addition to the new Figure 8:

Figure 8: Posterior distributions for H_0 for GRB060614, GRB150101B, GRB160821B, GRB050709, and their combined posteriors (note: without GW170817), are shown. The left column is the Kasen ²³ model and the right column is the Bulla ²⁴ model, with the top row corresponding to 0.1 mag errors and bottom row corresponding to 0.25 mag errors. The 1- and 2- σ regions determined by the “superluminal” motion measurement from the radio counterpart (blue) ¹⁷, Planck CMB (TT,TE,EE+lowP+lensing) (green) ³⁴ and SHoES Cepheid-SN distance ladder surveys (orange) ⁷ are also depicted as vertical bands.

Sincerely,
Michael W. Coughlin

Michael W Coughlin

REVIEWER COMMENTS

Reviewer #1 (Remarks to the Author):

Dear Authors,

I am happy to see the improvements in the paper which makes a stronger case for the publication. However there are some partially addressed issues before acceptance for publication. I understand that my scientific bias could cloud my view to see the pathological issues. Please clarify such cases.

“We restrict $0 < \theta_{\text{obs}} < 15^\circ$ for the sGRB analysis because the viewing angle is much closer to the polar axis than for GW170817\cite{TrRy2018}.”

>> Is this true for all sGRB? Do you expect them to be such for all sGRB that will be observed in the future.

“This means that our analysis is even more general since it indeed holds for all possible parameters, no numerical relativity results enter without our study; we add an additional note to the manuscript stating this fact.”

“This analysis shows that our flat priors cover both of the confirmed events so far, and in this sense, applying flat priors has the significant advantage that a larger range of the parameter space is covered. Further, it reduces systematic uncertainties since no numerical-relativity inferred relations have been applied.”

“This point is related to point 1a and we hope that our answers provided already a better insight. The models from Kasen et al. and Bulla et al. are agnostic to the specific binary systems and properties. That is, they are generic enough to be used to model both BNS and NSBH systems.”

>> I fail to emphasize the point clearly.

Is it possible that by taking all possible combinations of parameters, you are enforcing the standardization?

Since by taking a large number of samples in the unphysical region of parameter space, it is possible to get unrealistically low error. It might happen that if you include only the physically possible combinations, the scatter is too large to be accepted as a standardizable candle. Just a small demonstration would suffice where you take perhaps an equal number of samples but, in the physically possible region of parameter space.

A further question arises whether the standardization for the BNS system would be the same as the BH-NS system due to their different position in parameter space.

“Studying the predictions of two independent models with physical assumptions allows us to estimate the systematic uncertainties of our analysis.”

>> The systematic uncertainties also includes the unmodeled effects, is it possible to get an estimate of that? For example, by choosing a spherically symmetric model or a 2-component model of kilonova light curves, we might get an illusion of several special properties which might not be present in the actual system.

Since the field is still quite young in the literature, just a mention of this point should suffice.

“In addition, the impression that our results are more accurate than Planck or SHOES data is incorrect.”

>> I apologize for this confusion, Authors are correct. Thanks for pointing this out.

“We have added a section (Lightcurves) to the methods discussing the source of the data and remind that the r-band observations are used to standardize the SGRBs due to the sparsity of K-band data, and caveats about GRB060614’s source in particular; exactly as pointed out by the referee.”

>> The original objection was on the use of different bands for Hubble-Lemaitre constant and the derivation of standardization using models. This has not been addressed. In fact, it has become more confusing now because it’s not clear which band is used in Fig. 5. This has to be r-band in order for it to apply to sGRBs for which only r-band observation is of any use.

Reviewer #2 (Remarks to the Author):

The authors address my concerns in the previous report. Therefore, I recommend this article to be accepted.

UNIVERSITY OF MINNESOTA

Twin Cities Campus

School of Physics and Astronomy
College of Science and Engineering

Tate Laboratory of Physics
116 Church Street S.E.
Minneapolis, MN 55455-0112
612-624-7375
Fax: 612-624-4578
Website: www.physics.umn.edu

Dear Referees,

We are grateful for the constructive comments of the first referee and the proposed acceptance by the second. As for the first comments, we tried our best to incorporate the suggested changes and to provide sufficient answers to further improve the manuscript that we submit again for further evaluation.

Individual responses to all comments and suggestions are inline below.

Reviewer #1:

Dear Authors,

I am happy to see the improvements in the paper which makes a stronger case for the publication. However there are some partially addressed issues before acceptance for publication. I understand that my scientific bias could cloud my view to see the pathological issues. Please clarify such cases.

“We restrict $0 < \theta_{\text{obs}} < 15^\circ$ for the sGRB analysis because the viewing angle is much closer to the polar axis than for GW170817 [TrRy2018].”

>> Is this true for all sGRB? Do you expect them to be such for all sGRB that will be observed in the future.

We thank the referee for this question. We believe that this is a reasonable assumption, and that it will indeed apply in future sGRB events. Perhaps as the understanding of GRBs and the selection biases from GRB satellites improve, this assumption may need to be relaxed somewhat, especially considering that the literature doesn't seem to come to a complete consensus about opening angles. That being said, we turn to the work of Fong et. al., D'Avanzo, Saleem, and Beniamini et. al. to demonstrate that only mergers with small inclination angles are consistent with the observed sGRBs. As demonstrated in these articles, sGRB jet half-opening angles are likely much narrower than 15 degrees (around 0.1 str). We added the text: *“By observing the sGRB, we can assume that we are near or within the opening angle of the sGRB jet, which is taken to be less than 15° [cite{Fong:2013lba, DAv2015, Sal2020, BePe2018}].”* to the “Kilonova Analysis” section.

“This means that our analysis is even more general since it indeed holds for all possible parameters, no numerical relativity results enter without our study; we add an additional note to the manuscript stating this fact.”

“This analysis shows that our flat priors cover both of the confirmed events so far, and in this sense, applying flat priors has the significant advantage that a larger range of the parameter space is covered. Further, it reduces systematic uncertainties since no numerical-relativity inferred relations have been applied.”

“This point is related to point 1a and we hope that our answers provided already a better insight. The models from Kasen et al. and Bulla et al. are agnostic to the specific binary systems and properties. That is, they are generic enough to be used to model both BNS and NSBH systems.”

>> I fail to emphasize the point clearly.

Is it possible that by taking all possible combinations of parameters, you are enforcing the standardization? Since by taking a large number of samples in the unphysical region of parameter space, it is possible to get unrealistically low error. It might happen that if you include only the physically possible combinations, the scatter is too large to be accepted as a standardizable candle. Just a small demonstration would suffice where you take perhaps an equal number of samples but, in the physically possible region of parameter space.

We understand the concern of the referee, and to explore this, we have conducted further investigations and conclude that the effect of using broad, flat parameter distributions applied to either BNS or NSBH coalescences has minimal effect on the standardization. This validates our conservative approach in this work. We have added Figure 10 (see below) and the following text: *“When performing the fits, we rely on the assumption that the standardization is equally valid across the parameter space. Until there are further unambiguous kilonova detections, in particular those that also have GW counterparts, it will be difficult to create more strongly informed priors; for now, doing so would require assumptions on both the distribution of BNS and BHNS masses and an estimate of the NS EOS. It is not so obvious what unphysical regions there are yet, outside of likely very high ejecta masses, which are already excluded by our priors. To evaluate the effect this may have, Fig.~\ref{fig:fundamental} shows the performance of the fit of Eqs.~\eqref{eq:fit_inferred_bulla} compared to the Bulla\cite{Bul2019} model for both “Broad” and “Realistic” priors, i.e. those used in this analysis. Here, the “Broad” prior values are derived from the entire available grid, covering a range $-6 \leq \log_{10}(M_{\text{ej}}/M_{\odot}) \leq -0$, $15^{\circ} \leq \Phi \leq 75^{\circ}$, and $0 \leq \theta_{\text{obs}} \leq 90$. The “Realistic” prior values are derived from the prior range used in the sampling, $-3 \leq \log_{10}(M_{\text{ej}}/M_{\odot}) \leq -1$, $15^{\circ} \leq \Phi \leq 30^{\circ}$, and $0^{\circ} \leq \theta_{\text{obs}} \leq 15^{\circ}$, which was tuned for the sGRBs. In general, the estimated values from both versions of the fit are consistent with one another and the measured values from the model, and therefore, the effect on the standardization from this perspective is minimal. In the future, if there are regions which are deemed to be disfavored, the priors should be updated to reflect this.”*

A further question arises whether the standardization for the BNS system would be the same as the BH-NS system due to their different position in parameter space.

We thank the referee for raising this question. We agree that the standardization for BNS systems could be different from that for BH-NS systems. However, it is currently unclear what exact regions of the parameter space the two systems occupy and whether these are distinct or overlap. Future kilonovae (well characterized in multiple bands with photometry at early times) will help address this question and pin down the ejecta geometry and ejecta mass ratio for BNS and BH-NS, helping set priors on the ϕ and M_{ej} parameters of our grid.

To make this point clear, we added the following statement: *“As for assumptions about the ejecta geometry, the current version is generic enough such that it is applicable to both BNS and BHNS cases. However, an improved geometry would likely need to be split into a BNS and BHNS case, which would mean that the standardization for BNS systems could be different from the one for BHNS systems. However, it is currently difficult to assess what regions of the parameter space are favoured for each system and whether these are distinct or overlap. The detection of more kilonovae in the future will help pin down the ejecta geometry and ejecta mass ratio for BNS and BHNS, allowing us to update our priors on ϕ and M_{ej} and investigate the possibility of different standardizations.”*

“Studying the predictions of two independent models with physical assumptions allows us to estimate the systematic uncertainties of our analysis.”

>> The systematic uncertainties also includes the unmodeled effects, is it possible to get an estimate of that? For example, by choosing a spherically symmetric model or a 2-component model of kilonova light curves, we might get an illusion of several special properties which might not be present in the actual system. Since the field is still quite young in the literature, just a mention of this point should suffice.

We thank the referee for this recommendation. We added the line *“More specifically, some of these physical assumptions (e.g. spherical or axial symmetry) may give the false impression of special kilonovae properties. Doing this study with multiple different models is therefore critical to reveal such systematics.”* to the end of the “Kilonova Analysis” section. The authors are preparing a manuscript which hopes to address this question more directly. We are currently investigating with an intensive injection campaign the difference between surrogates created via the different studies of Kasen et. al., 2017 and Bulla, 2019. However, this study is, due to its large computational footprint, still ongoing.

“We have added a section (Lightcurves) to the methods discussing the source of the data and remind that the r-band observations are used to standardize the SGRBs due to the sparsity of K-band data, and caveats about GRB060614’s source in particular; exactly as pointed out by the referee.”

>> The original objection was on the use of different bands for Hubble-Lemaitre constant and the derivation of standardization using models. This has not been addressed. In fact, it has become more confusing now because it’s not clear which band is used in Fig. 5. This has to be r-band in order for it to apply to sGRBs for which only r-band observation is of any use.

We thank the referee for pointing out this apparent inconsistency. We now write in the text: *“We choose r -band for the GW170817 analysis when analyzed on its own, given that the transient was observable for the longest in the near-infrared; we chose r -band for the sGRB analyses, including in the standardization of GW170817 in the magnitude differences computed below, as it was the band most commonly imaged between the various sGRBs.”* and the caption of Figure 5 now reads, *“... varying the opening and viewing angles of the employed simulation set, in r -band.”*

Reviewer #2 (Remarks to the Author):

The authors address my concerns in the previous report. Therefore, I recommend this article to be accepted.

The authors thank the reviewer for the positive feedback and again for their helpful suggestions during the previous round of comments.

Thank you,

Michael W. Coughlin for the authors

REVIEWERS' COMMENTS:

Reviewer #1 (Remarks to the Author):

Dear Authors,

I believe the raised concern has been addressed to sufficient accuracy possible in the field. I propose the paper for acceptance.

Thank you for your patience.

UNIVERSITY OF MINNESOTA

Twin Cities Campus

*School of Physics and Astronomy
College of Science and Engineering*

*Tate Laboratory of Physics
116 Church Street S.E.
Minneapolis, MN 55455-0112
612-624-7375
Fax: 612-624-4578
Website: www.physics.umn.edu*

Dear Referees,

Reviewer #1 (Remarks to the Author):

Dear Authors,

I believe the raised concern has been addressed to sufficient accuracy possible in the field. I propose the paper for acceptance.

Thank you for your patience.

We are grateful for the constructive comments of the referees throughout this process, and thank them again for their helpful suggestions during the previous rounds of comments.

Thank you,

Michael W. Coughlin
for the authors